# Mechanical Behavior of Brick Masonry in an Acidic Atmospheric Environment

**DOI:** 10.3390/ma12172694

**Published:** 2019-08-23

**Authors:** Shansuo Zheng, Lihua Niu, Pei Pei, Jinqi Dong

**Affiliations:** Department of Civil Engineering, Xi’an University of Architecture and Technology, Xi’an 710055, China

**Keywords:** brick masonry, acid rain corrosion, shear bond strength, compressive strength, mathematical degradation model

## Abstract

In order to evaluate the deterioration regularity for the mechanical properties of brick masonry due to acid rain corrosion, a series of mechanical property tests for mortars, bricks, shear prisms, and compressive prisms after acid rain corrosion were conducted. The apparent morphology and the compressive strength of the masonry materials (cement mortar, cement-lime mortar, cement-fly ash mortar, and brick), the shear behavior of the masonry, and the compression behavior of the masonry were analyzed. The resistance of acid rain corrosion for the cement-lime mortar prisms was the worst, and the incorporation of fly ash into the cement mortar did not improve the acid rain corrosion resistance. The effect of the acid rain corrosion damage on the mechanical properties for the brick was significant. With an increasing number of acid rain corrosion cycles, the compressive strength of the mortar prisms, and the shear and compressive strengths of the brick masonry first increased and then decreased. The peak stress first increased and then decreased whereas the peak strain gradually increased. The slope of the stress-strain curve for the compression prisms gradually decreased. Furthermore, a mathematical degradation model for the compressive strength of the masonry material (cement mortar, cement-lime mortar, cement-fly ash mortar, and brick), as well as the shear strength attenuation model and the compressive strength attenuation model of brick masonry after acid rain corrosion were proposed.

## 1. Introduction

Masonry structures and architectural heritage undergo several decay processes due to exposure to aggressive environmental conditions that threaten its durability and mechanical properties [1]. Recently, atmospheric pollution (acid deposition) on masonry materials has been recognized as one of the most important and common reasons of decay endangering the built heritage [2,3,4]. High acid rain concentration caused masonry materials to dissolve and form harmful salts, which leads to a significant reduction in the mechanical properties of masonry material and the structural service life [5,6]. At the present, there are a large number of masonry structures and cultural heritage of brick masonry worldwide, and they have been exposed to an acidic atmospheric environment for a long time. Therefore, it is necessary to evaluate the degradation of mechanical properties in the acid rain area, which allows more effective restoration works.

For the past few years, the impacts of acid deposition on the weathering of masonry building material (stone) have been discovered and examined. Many researchers have focused on the identification of the different processes responsible for stone dissolution [7], the influences of acid deposition on the decay of stones, and the quantitative relationships between climatic variables [8,9]. The chemical analysis of the run-off solutions, direct measurement of recession, the surface modifications of building stone with an electron scanning microscope, and the X-ray diffraction analysis of stone in response to environmental aggressiveness were studied with consideration of the important role of stone characteristics, weight loss measurements, and the forecast of stone [10,11,12]. Furthermore, the correlation between stone microstructural characteristics and material degradation was investigated [13].

Additionally, several aspects of mortar and bricks under simulated acid rain conditions were evaluated. The change in mineralogical composition and chemical behavior, as well as the strength, phase, and internal structure of the mortar, were studied in a previous publication [14,15,16,17]. The effects of acid rain corrosion on the appearance, and the quality of masonry mortars, were studied [18]. The effect of low-calcium fly ash on mortar strength under simulated acid rain conditions was studied, and the flexural strength of cement-fly ash mortar specimens was lower than that of pure cement mortar specimens after acid rain corrosion [19,20]. J.A. Larbi described the integrated microscopic method for diagnosing the causes and the degradation extent of fired clay brick masonry under atmospheric environment [21]. The damage mechanisms of building materials (stones, mortars) after acid rain corrosion were described. The degradation degree of mortar was higher than that of stone [22,23].

The main ion components in acid rain are SO_4_^2−^, NO^3−^, Ca^2+^, NH_4_^+^, Mg^+^, H^+^, and H^+^. SO_4_^2−^ leads to serious corrosion damage to mortar [17,24,25]. The acid rain corrosion mechanism of masonry material is a complex process, including acid corrosion (dissolved corrosion) and salt corrosion (expansion corrosion) in precipitation. 

In conclusion, many research studies have been devoted to chemical components, density, porosity, vapor permeability, water absorption, and the acid rain corrosion mechanism in materials (such as natural stone, brick, mortar, and concrete). However, the effect of an acidic atmospheric environment on the mechanical performance of masonry has not been fully elucidated. Therefore, the aim of this paper is to evaluate the mechanical behavior of a masonry building in an acidic atmospheric environment, including the mechanical characteristics of the masonry unit, mortar, unit-mortar bond, and masonry prism. Thus, simulated acid rain corrosion tests and mechanical performance tests on standard mortar prisms, bricks, masonry shear specimens, and masonry compression specimens were conducted. Furthermore, mathematical strength degradation models for masonry were established.

## 2. Materials and Experimental Methods

### 2.1. Materials and Specimen Preparation

The raw materials used in this work included cement, lime paste, first grade fly ash, medium sand, and a water reducing agent. The properties and the resultant grading of the P.O 32.5 ordinary Portland cement are shown in Table 1. Water from an urban water supply, and hard and good gradation II Region river sand was used, for which the fineness modulus did not exceed 2.75% and the percent of mud did not exceed 1.5%. The properties of fly ash are shown in Table 2. The lime paste used in the research testing was produced by a building material company in Xi’an. Its main component was Ca(OH)_2_ and it contained a small amount of CaO.

The properties of the brick are summarized in Table 3. The dimension of the brick was 240 × 115 × 53 mm^3^ (length × width × height). Cement mortar, cement-lime mortar, and cement-fly ash mortar were prepared because they were widely used in Southwest China where acid rain pollution was quite severe. The mortar specimen dimensions were 70.7 × 70.7 × 70.7 mm^3^ [26]. The mix ratios of the mortars are shown in Table 4.

In order to quantify the evolution of the mechanical properties, three measurements were made: the compressive strength of the mortar, the shear behavior, and the compression behavior of the masonry. The specimen geometric sizes and the number of specimens used for each test and testing condition are summarized in Table 5 and Figure 1. Three different types of mortar prisms, that is, a cement mortar prism (CEM), a cement-lime mortar prism (CEM-LIM), and a cement-fly ash mortar prism (CEM-FLY), were prepared in order to obtain a variety of physical properties, which allows different acid rain resistances. Seven groups of mortar prisms were prepared for each type of mortar and brick, corresponding to different acid corrosion cycles (0 cycle, 50 cycles, 100 cycles, 150 cycles, 200 cycles, 250 cycles, and 300 cycles). In order to assess the shear bond strength between mortar and brick under acid rain corrosion and to reveal the deterioration regularity of the shear-bond strength caused by acid rain corrosion damage, 28 shear specimens with different acid corrosion cycles were tested under axial compression. The specimens were divided into four groups, which were called KJMA, KJMB, KJMC, and KJMD(KJM stand for the shear masonry in Chinese) corresponding to 0, 100, 200, and 300 designed acid corrosion cycles, respectively. Seven specimens were tested for each group, according to the minimum required by GB/T50129-2011 [27]. In order to obtain the basic mechanical properties of the masonry material under acid rain corrosion, 32 brick masonry compressive specimens with different acid corrosion cycles were designed. The specimens were divided into four groups, which were called KYMA, KYMB, KYMC, and KYMD (KYM stand for the compressive masonry in Chinese), corresponding to 0, 40, 80, and 120 designed acid corrosion cycles, respectively. Eight specimens were tested for each group, according to the minimum required by GB/T50129-2011 [27]. For each type of specimen, the mortar compressive strength rating was M10 and a uniform mortar thickness of 10 mm was adopted for the joint mortar.

### 2.2. Laboratory Exposure Chambers

In order to simulate an acidic atmospheric environment, it was decided to simulate acid rain by using sulfuric acid and nitric acid. The reason can be explained as follows. Various sulfurous pollutants and nitrous pollutants can be oxidized and converted to sulfuric acid and nitric acid with higher acidity. Sulfuric acid and nitric acid are still relevant in many zones, particularly in developing countries such as China [28]. To better simulate the strong acid rain and shorten the test period, much research was reported on with regard to accelerating the concrete corrosion in an acid rain environment, where the pH value decreases and the concentration of acid radical ions increases [29]. Therefore, an acid solution of pH = 3.5 was prepared, in which the major constituents were H_2_SO_4_ and HNO_3_ with a proportion of 9:1. The specimens were cured for 28 days in a natural environment, and they were placed into the environment chamber to reach the desired levels of corrosion damage. Specimens in the laboratory exposure chambers are shown in Figure 2. The acid rain corrosion rule is shown in Figure 3.

### 2.3. Loading Scheme

#### 2.3.1. Compressive Test on a Mortar Prism and a Brick

Figure 4 shows the compression test of mortar prisms. According to the Chinese standards JGJ/T 70-2009 [30], the appearance damage and geometry size of the mortar prism were recorded before the mortar prism was subjected to the compression test. Then, the mortar prisms were placed on the lower press plate of a universal axial testing machine. To ensure a uniform force of the mortar prisms, the center of the mortar prisms was aligned with the center of the upper and lower press plates of the universal axial testing machine and the surface of mortar prisms was placed parallel to the contact surface of the upper and lower pressure plates. The loading procedure was load controlled monotonically at a loading rate of 1.5 kN/s.

Figure 5 shows the compression test of the brick. The compressive strengths of bricks subjected to different acid rain corrosion cycles were determined according to Chinese standards [27]. The length and the width of two bonded surfaces of brick and geometry size of the brick were recorded. Then the compression experiment was conducted. The loading speed was controlled to 4 kN/s so that the brick was evenly loaded.

#### 2.3.2. Direct Shear Test

Figure 6 shows the shear test loading of the specimen. The direct shear test was conducted on the WAW-1000 universal axial testing machine of the structural engineering and seismic laboratory of the Xi’an University of Architecture and Technology. The shear test was performed in accordance with the Chinese standards GB/T 50129-2011 [27], which can be stated as follows. (1) The size of the sheared surface was measured and the measuring accuracy was 1 mm. (2) The masonry shear specimen was placed on the bar-shaped pressure plate of the testing machine and the centerline of the specimen coincided with the axis of the upper and lower plate of the test machine. To ensure that the upper and lower press plates was in close contact with the masonry shear specimen, a 10-mm thick hard rubber was placed between the specimen and the bar-shaped pressure plate. (3) A direct shear test was carried out by a uniform and continuous load. The load speed was controlled by breaking the specimen within 1 to 3 minutes. When the sheared surface was damaged, the specimen was considered destroyed.

#### 2.3.3. Uniaxial Compressive Test

Figure 7 shows the test loading device of the masonry compressive specimen. Prior to the test, strain gages were attached to the compressive specimen vertically and horizontally. The gauge length of the horizontally attached strain gage was 265 mm, and the gauge length of the vertically attached strain gage was 325 mm. In order to check the sensitivity of the instrument and the firmness of the installation, a 5% estimated failure load was first applied to the compressive specimen. Then 5% to 20% of the estimated failure load was applied to the specimen and the pre-load was repeated (three to five times) in order to adjust the axial deformation of the wide sides of the compressive specimen. The consecutive loading method was adopted and 10% of the estimated failure load was applied per load step. In addition, the uniform acceleration was completed within 1 to 1.5 minutes. The compressive specimen declared damage when the load reached the ultimate load.

## 3. Results and Discussion

The results of apparent morphology and compressive strength of mortars subjected to different acid rain corrosion cycles have been presented in the subsequent sections. The effect of acid rain corrosion on the failure process and shear strengths of brick-mortar were discussed. Furthermore, the failure process and pattern, compression strengths, and stress–strain characteristics of masonry prisms were addressed.

### 3.1. Mortar and Brick

#### 3.1.1. Apparent Morphology

Figure 8 shows the apparent morphology of a mortar prism after acid rain corrosion. Figure 8a shows an uncorroded mortar prism as a comparative test. In the initial stage of acid rain corrosion, the color of the mortar prism surface changed from gray to dark. The crystal and sanding phenomena appeared on the mortar prism surface, as shown in Figure 8b. In the middle stage of acid rain corrosion, the peeling appearance of the mortar prism surface and the color of the mortar prism became lighter, as shown in Figure 8c. At the end stage of the acid rain corrosion, the crumbling and blistering appearance of the mortar prism surface became more and more serious, and the appearance of flaking and disintegration could be observed in some places, as shown in Figure 8d.

The interpretation of the above observations may be summarized as follows. During the initial stage of acid rain corrosion, the white crystal on the mortar prism surface was caused by the chemical reaction of Ca(OH)_2_, SO_4_^2−^ and H_2_O. At the later stage of the acid rain corrosion, H^+^ and Ca(OH)_2_ on the surface of the mortar prisms reacted to form Ca^2+^, which caused the alkalinity of the mortar to drastically decrease. The internal hydration calcium silicate and the calcium aluminate hydration lost their stability and accelerated their hydrolysis, destroying the gel structure of the mortar, which led to the mortar occurring in the layer-by-layer corrosion damage from the outside to the inside [18].

Figure 9 shows the apparent morphology of brick subjected to different acid rain corrosion cycles, which could be divided into three stages. Figure 9a shows uncorroded brick as a comparative test. In the initial stage of the acid rain corrosion, the surface color of bricks became deeper and darker, as shown in Figure 9b. In the middle stage of the acid rain corrosion, powdery substances formed on the surface of the brick, as shown in Figure 9c. At the end stage of the acid rain corrosion, the surface of brick appeared to be “skinning” and the phenomenon of “open mouth” in the corners was clear, as shown in Figure 9d.

#### 3.1.2. Compressive Strength

Compressive strength is a basic mechanical properties index of building materials. Studying the compressive strength of mortar prisms and bricks after acid rain corrosion is of great significance. The compressive strength (f) of a standard test cube of specimens can be calculated from Equation (1). To facilitate the change law of the compressive strength for a mortar prism and a brick with different corrosion degrees, the loss rate of compressive strength ($Q_{n}$) is defined as shown in Equation (2).
(1)$$f=\frac{N}{A}$$
(2)$$Q_{n}=\frac{f_{c,0}-f_{c,n}}{f_{c,0}}\times 100\%$$
where, N is the compressive load, kN. A is the cross-section of the specimen, mm^2^. $f_{c,0}$ is the average compressive strength of uncorroded mortar prism, MPa. $f_{c,n}$ is the average compressive strength of the mortar prism subjected to *n* acid rain corrosion cycles, MPa.

Table 6 shows the compressive strengths (mean of six specimens) of three typical mortar prisms (cement mortar, cement-lime mortar, and cement-fly ash mortar) and bricks subjected to different corrosion cycles, including the standard deviation and the strength loss rate. It can be observed that the compressive strength of the mortar prism first increased and then decreased as the number of acid rain corrosion cycles increased (which was in good accordance with other studies on concrete) [31]. According to the corrosion mechanisms of cement concrete [28,32], the reason for this phenomenon can be stated as follows: The SO_4_^2−^ ions in the corrosion solution penetrated into the micro-pores in the mortar prism, producing expanded CaSO_4_·2H_2_O crystals. In the initial stage of acid rain corrosion, CaSO_4_·2H_2_O filled in the surface and pores of the mortar prism and the internal compactness increased, which increased the compressive strength of the mortar prism. At the same time, the H^+^ ions in the corrosion solution reacted with the Ca(OH)_2_ on the surface of the mortar prism, causing the surface of the mortar prism to undergo corrosion damage, which resulted in the decrease in the compressive strength of the mortar prism. In the initial stage of corrosion, the effect of SO_4_^2−^ ions on the compressive strength of the mortar was higher than that of the H^+^ ions, which resulted in an increase in the compressive strength of the mortar prism. With the increasing number of corrosion cycles, the damage caused by the corrosion effect of the H^+^ ions accumulated. The expansion corrosion product (hydrated calcium sulphoaluminate) produced by the SO_4_^2−^ ions gradually increased. The surface stress and the internal stress of the mortar prism increased due to the increasing volume of the corrosion products, which leads to micro-cracks in the mortar. At the same time, the generated micro-cracks made it easier for hydrogen ions and sulfate ions to intrude into the mortar, which resulted in further deterioration of the mechanical properties of the mortar.

The compressive strengths of the cement-lime mortar prism, the cement mortar prism, and the cement-fly ash mortar prism subjected to 100 acid rain corrosion cycles increased by 10.42%, 9.78%, and 14.81%, respectively, which indicates that the strength of the cement-fly ash mortar increased rapidly in the early stage of corrosion. The interpretation of this result may be summarized as follows. The pozzolanic effect and the micro-filling effect of the cement-fly ash mortar prism improved the uniformity of the material gradation, which resulted in more ettringite gel and thaumasite in the hydration process of the cementitious material [33,34]. In addition, the CaSO_4_·2H_2_O crystals formed by the cement-fly ash mortar subjected to acid rain corrosion were more than pure cement mortar (which has been confirmed by other authors) [19]. Furthermore, the ettringite gel and the CaSO_4_·2H_2_O crystals could improve the compactness of the mortar prism. The compressive strengths of the cement-lime mortar prism, cement mortar prism, and cement-fly ash mortar prism subjected to 300 acid rain corrosion cycles were reduced by 24.79%, 15.30%, and 19.01%, respectively, which indicated that the cement-lime mortar prism had the worst corrosion resistance. The excess gypsum constituent of the cement-lime mortar reacted with the tricalcium aluminate to generate a large amount of hydration calcium sulpho-aluminate (a volume increase of approximately 150%), which destroyed the microscopic pore structure of the mortar. This resulted in a reduction of the compressive strength.

The compressive strength of the brick decreased from 16.37 MPa to 13.06 MPa (20.22%) after 300 acid rain corrosion cycles, which indicated that the effect of the acid rain corrosion damage on the mechanical properties for the brick was significant. The medium compressive strength of the brick was higher than that of the mortar under the same corrosion cycles, which indicated that the masonry shear strength was limited by the shear strength of mortar.

### 3.2. Shear Behavior of the Masonry

#### 3.2.1. Failure Process and Pattern

Figure 10 illustrates the final damage state of the shear specimens subjected to 0 (KJMA), 100 (KJMB), 200 (KJMC), and 300 (KJMD) acid rain corrosion cycles. The failure processes of the specimens were essentially the same, and all the shear specimens showed brittle failures with monotonic vertical loading. At the initial stage of loading, the overall shear resistances of the masonry shear specimens were not affected by acid rain corrosion. When the vertical load reached the ultimate shear load, the specimens underwent sudden brittle failure along the sheared surface without any symptoms.

By comparing the final damage states of the shear specimens with different amounts of acid rain corrosion, it can be observed that the shear failure surface was no longer flat and it became more irregular as the number of acid corrosion cycles increased. The reason for this can be explained as follows. As the number of acid corrosion cycles increased, the compressive strength of the mortar decreased and the gelled material in the outer cement-based material was lost. Then, the sulfate crystals formed an intumescent substance and micro-cracks appeared on the brick–mortar interface under the expansion pressure. When the vertical load reached the ultimate shear load, a stress concentrate phenomenon occurred at the brick-mortar interface and the micro-cracks further propagated. Hence, irregular splitting damage formed in the brick-mortar interface under the action of the external force.

Figure 11 illustrates the main failure patterns of the shear specimens in the direct shear test. The failure pattern of the shear specimen was manifested as a single shear or a double shear. Referring to the single–shear modes, the failure occurred with the separation of the mortar from the brick due to the weakness of the brick–mortar interface. When the double–shear failure occurred, the crack first appeared on the bond interface with lower shear strength. With the load increasing, the first crack developed to a certain degree, the other shear interface appeared to crack one after another, and the time interval between the two cracks was short. In particular, due to the close compressive strength of the masonry mortar and brick, the mortar at the adhesive interface of some shear specimens was destroyed in the bed joint along a 45° angle under the action of the main tensile stress.

#### 3.2.2. Shear Strength of the Masonry

The results of the direct shear test of the 28 masonry shear specimens after different acid rain corrosion cycles are shown in Table 7, while the shear strength ($f_{\mathrm{vm}}$) of the individual specimens can be deduced from the following equation.
(3)$$f_{\mathrm{vm}}=N_{\mathrm{vu}}/2A$$
where $N_{\mathrm{vu}}$ is the ultimate shear load, kN. A is the cross-section area of sheared surface, mm^2^.

For each specimen, the ultimate load, the shear strength, the mean shear bond strength, and the failure mode (single shear or double) were illustrated in Table 7. It can be seen that the shear strength of the masonry specimen first increased and then decreased as the number of acid rain corrosion cycles increased. Compared with the strength of an uncorroded specimen, the shear strengths of the specimens increased by 2.4% and 1.2% after 100 and 200 acid rain corrosion cycles. The reason for this was that the corrosion solution penetrated into the mortar and inside of bricks, and then the Ca^2+^, Al^3+^, and other reactions generated sulphoaluminate and ettringite crystals, which resulted in an increase in the densities of the bricks and mortar. The higher shear strength can also be attributed to the accelerated carbonation caused by favorable conditions in the climatic chamber. After 300 acid rain corrosion cycles, the shear strength of the brick masonry specimens built by the cement-lime mortar decreased by 13.1%. The reason for this was the development of micro-cracks for the mortar and the reduction in the shear bond strength due to the weakness of the brick-mortar interface.

The masonry shear strength values dropped with the decreasing compressive strength of the mortar, but the lower rate was not significant when the compressive strength of the mortar was low. The compressive strength of the cement-lime mortar subjected to 100 acid rain corrosion cycles increased by 10.4%, while the shear strength of the masonry subjected to 100 acid rain corrosion cycles grew by 2.4%. The compressive strength of the cement-lime mortar subjected to 300 acid rain corrosion cycles was reduced by 24.8%, while the shear strength of the masonry subjected to 100 acid rain corrosion cycles was reduced by 28.4%.

### 3.3. Compression Behavior of Masonry

#### 3.3.1. Failure Process and Pattern

Figure 12 illustrates the final damage state of the compression specimens subjected to 0 (KYMA), 100 (KYMB), 200 (KYMC), and 300 (KYMD) acid rain corrosion cycles. The failure processes of the specimens were essentially the same and all specimens experienced initial cracking, crack development, and failure stages. In the initial cracking stages, the stress increased uniformly with the growth of the strain and these factors were approximately linearly related, which indicates that the specimens were in the elastic stage. When the vertical load reached 50% to 70% of the ultimate load, the first batch of vertical cracks was observed in the specimen. In the crack development stages, the initial crack gradually extended downward along the longitudinal direction with the increase of the vertical load. When the load reached 80% to 90% of the ultimate load, the primary crack developed into a main vertical crack and there were several vertical micro-cracks parallel to the main vertical crack [35]. In the failure stages, the width of the main vertical crack gradually increased with the further increase of the vertical load, some bricks were crushed, and the lateral strain of the specimen increased rapidly. When the load reached the ultimate load, the specimens were divided into several small prisms by the penetrating crack and failure was ultimately pronounced.

The differences in the failure process of the specimens subjected to 0 (KYMA), 100 (KYMB), 200 (KYMC), and 300 (KYMD) acid rain corrosion cycles can be summarized as follows: (1) Compared with the group KYMA and KYMB specimens, the number of vertical cracks in groups KYMC and KYMD were significantly larger in quantity, the distribution of vertical cracks in groups KYMC and KYMD was more dispersed, the average cumulative crack widths in groups KYMC and KYMD was wider, and the main vertical crack lengths on the narrow side surfaces in groups KYMC and KYMD was longer (the maximum length of the main vertical cracks of group KYMD was 693 mm whereas the maximum length of the main vertical cracks of group KYMA was 424 mm). This phenomenon showed that the specimens were damaged because acid rain corrosion and micro-cracks were more likely to take place when the specimens were subjected to a vertical load. (2) The surface spalling phenomenon occurred on the edges of the surfaces in groups KYMC and KYMD and the damage degree of the compression specimens increased with the number of acid rain corrosion cycles. This phenomenon was ascribed to the vertical micro-cracks caused by acid rain corrosion. The vertical micro-cracks developed and penetrated as the vertical load increased. Then one layer of the edge for the compression specimen was easily “cut away.” (3) Two vertical cracks appeared at the bottom 1/4 of the height in group KYMD (Figure 12) and partial crushing occurred at the feet of the specimen (Figure 13). The reason for this phenomenon was that the simulated corrosion solution slowly flowed down along the specimen and stacked at the bottom. Hence, the acid corrosion degree at the bottom of the specimen was the most serious, which resulted in a weak part at the bottom of the specimen. (4) As the number of acid rain corrosion cycles increased, the initial crack load of the compression specimen gradually decreased. This means that the time for the occurrence of the initial crack gradually advanced.

#### 3.3.2. Compression Strength of Masonry

The measured compressive strength ($f_{\mathrm{cm}}$) of the individual specimen can be deduced from the following equation.
(4)$$f_{\mathrm{cm}}=N_{\mathrm{cu}}/A$$
where $N_{\mathrm{cu}}$ is the ultimate load of the compressive specimen, kN. A is the cross-section area of compressive specimen, mm^2^, which was calculated according to the average width and average thickness of the specimen.

According to the specification [27], the calculated values of masonry compressive strength ($f_{m}^{c}$) is calculated by Equation (5).
(5)$$f_{m}^{c}=k_{1}f_{1}^{\alpha }\left(1+0.07f_{2}\right)k_{2}$$
where $f_{1}$ is the average compressive strength of block, MPa, $f_{2}$ is the average compressive strength of mortar, MPa, $k_{1}$ is a parameter related to the type of block and construction method, α is a parameter related to the height of the block, and $k_{2}$ is a correction coefficient of compressive strength of mortar. The parameters $k_{1}=0.78$, α=0.5, and $k_{2}=1.0$ when calculating the average compressive strength of the compressive specimen.

Table 8 presents the characteristic values of compression performance, which included the crack load ($N_{\mathrm{cr}}$), ultimate load ($N_{\mathrm{cu}}$), measured compressive strength ($f_{\mathrm{cm}}$), and calculated compressive strength ($f_{m}^{c}$) of the specimens subjected to different acid rain corrosion cycles under vertical loading. The relative compressive strength (RCS1, ratio of measured compressive strength of specimens with and without acid rain corrosion, RCS2, ratio of calculated compressive strength of specimens with and without acid rain corrosion) were also adopted in Table 8.

Based on Table 8, the following observations can be made. (1) The average measured compressive strength of specimens increased slightly at first and then decreased as the number of acid rain corrosion cycles increased. The deteriorated regularity of compressive strength for masonry caused by acid corrosion was consistent with the acid corrosion damage observed in the masonry materials and shear specimen. (2) Compared with the un-corrosion specimen, the compressive strength of the specimen subjected to 100 acid rain corrosion cycles increased by 2.6%. After 300 acid rain corrosion cycles, the measured compressive strength of the masonry specimens built by cement-lime mortar decreased by 29%, which indicated that acid rain corrosion had great influence on the compressive strength of brick masonry. (3) After 300 acid rain corrosion cycles, RCS1 decreased from 1.0 to 0.706 (29%), and RCS1 decreased from 1.0 to 0.8 (20%). This finding indicated that the calculation results do not agree significantly with the experimental results because the effect of acid rain corrosion was not considered in Equation (11). Therefore, it is necessary to propose the degradation model of compressive strength.

#### 3.3.3. Stress–Strain Characteristics

According to the literature [36], an effective model on evaluating the stress–strain behavior of compression specimen subjected to different acid rain corrosion cycles was given, as illustrated in Equation (6). The different acid rain corrosion cycles were normalized and the fitting stress–strain curves were shown in Figure 14.
(6)$$\frac{\sigma }{f_{m}}=A\frac{\varepsilon }{\varepsilon_{m}}-B{\left(\frac{\varepsilon }{\varepsilon_{m}}\right)}^{C}$$
where $f_{m}$ is peak stress, $\varepsilon_{m}$ is the strain corresponding to peak stress, and A,B,C are fitting parameters.

Table 9 summarizes the parameters of the regression stress-strain curves for the compression specimens subjected to different acid corrosion cycles. As can be seen from Table 9, the average correlation coefficient $R^{2}=0.97$ of the stress—strain curve under different acid rain corrosion cycles was relatively high, which confirmed the good quality and repeatability of the results.

Figure 15 illustrates the compression stress versus the longitudinal strain for a compression specimen under different acid corrosion cycles. The observations made based on Figure 15 is as follows. (1) The tendency of these curves was essentially the same. Compared to the uncorroded specimen, the compressive specimens that were subjected to 100 acid rain corrosion cycles exhibited slightly increased in term of the peak stress, which was ascribed to chemical adsorption. With the number of acid rain corrosion cycles continuously increasing, the peak stress gradually decreased and the decreased degree was high. (2) The strain corresponding to the peak stress gradually increased with the increasing number of acid corrosion cycles. (3) The slope of the stress-strain curve for the compression specimen gradually decreased with the increasing number of acid corrosion cycles, which indicated that the elastic modulus of the compression specimen gradually decreased and the ability of the specimen to resist deformation gradually lowered. The above phenomena indicate that acid rain corrosion caused a significant decrease in the masonry mechanical properties.

Table 10 shows the elastic modulus of the specimen. According to the specification [27] and the stress-strain curve, the secant modulus at $\sigma =0.4f_{\mathrm{cm}}$ is taken as the elastic modulus of the specimen. It can be seen from Table 10 that the decline trend of the elastic modulus for brick masonry after different acid rain corrosion cycles was clear and the elastic modulus of the specimen subjected to 300 acid rain corrosion cycles decreased by 42%.

## 4. Degradation Model for Compressive Strength

This section provides an overview of the proposed models that were established to describe the test results of the mechanical properties for brick masonry subjected to different acid rain corrosion cycles. Through the comparison and statistical analysis of test data, it can be seen that the compressive strength of the cement mortar prism, cement-lime mortar prism, cement-fly ash mortar prism, brick unit, and acid rain corrosion cycle approximated a quadratic function relationship. The variations of the normalized compressive strength compared to their initial values are plotted in Figure 16. The analytical expressions of the compressive strength attenuation model for mortars and brick units are given in Equations (7) to (10).

Cement-lime mortar:(7)$$f(n)/f=1.002+6.611\times 10^{-4}n-5.511n^{2}$$

Cement mortar:(8)$$\mathrm{Cement}\;\mathrm{mortar}\;f(n)/f=1.000+0.001n-6.397\times 10^{-6}n^{2}$$

Cement-flyash mortar:(9)$$\mathrm{Cement}\text{-}\mathrm{flyash}\;\mathrm{mortar}\;f(n)/f=0.999+0.002n-9.796\times 10^{-6}n^{2}$$

Brick unit:(10)$$\mathrm{Brick}\;\mathrm{unit}\;f(n)/f=0.998-2.847\times 10^{-4}n-1.306\times 10^{-6}n^{2}$$
where f(n) is the compressive strength of the mortar prism and the brick subjected to *n* acid rain corrosion cycles. f is the compressive strengths of un-corrosion mortar prism and brick.

Through the comparison and statistical analysis of test data, it can be seen that the shear strength and compressive strength of masonry gradually decreased with the increasing number of acid rain corrosion cycles. The variations of the normalized shear strength and compressive strength compared to their initial values are plotted in Figure 17 and Figure 18. The evolution appeared to be quadratic with the number of acid rain corrosion cycles. Therefore, the shear strength attenuation model (Equation (11)) and compressive strength attenuation model (Equation (12)) of brick masonry considering the effect of acid rain corrosion cycles were proposed.
(11)$$\frac{f_{\mathrm{vm}}(n)}{f_{\mathrm{vm}}}=1.008+3.297\times 10^{-4}n-4.451\times 10^{-6}n^{2}$$
(12)$$\frac{f_{\mathrm{cm}}(n)}{f_{\mathrm{cm}}}=1.002+6.611\times 10^{-4}n-5.511\times 10^{-6}n^{2}$$
where, $f_{\mathrm{vm}}(n)$ is the shear strength of masonry subjected to *n* acid rain corrosion cycles, $f_{\mathrm{vm}}$ is the shear strength of the un-corrosion masonry, $f_{\mathrm{cm}}(n)$ is the compressive strength of masonry subjected to *n* acid rain corrosion cycles, and $f_{\mathrm{cm}}$ is the compressive strength of un-corrosion masonry.

## 5. Damage Constitutive Relationship Model

(1) The initial state of a specimen after maintenance is regarded as the first damage state, and the damage state after acid rain corrosion is regarded as the second damage state. Based on the strain equivalence principle, the following relationship can be obtained.
(13)$$D_{n}=(A_{0}-A_{n})/A_{0}$$
(14)$$\sigma_{0}A_{0}=\sigma_{n}A_{n}$$
(15)$$\varepsilon =\sigma_{0}/E_{n}=\sigma_{n}/E_{0}$$
where $\sigma_{0}$ and $\sigma_{n}$ are the effective stresses in the initial damage state and, after *n* acid rain corrosion cycles, respectively, $A_{0}$ is the cross-sectional area in the initial damage state and, after *n* acid rain corrosion cycles, respectively, $E_{0}$ and $E_{n}$ are the elastic moduli in the initial damage state and, after *n* acid rain corrosion cycles, respectively, and $D_{n}$ is the damage variable after *n* acid rain corrosion cycles.

Based on Equations (13) to (15), the relationship between the elastic modulus of the initial damage and the acid rain corrosion damage can be illustrated by Equation (16), and the acid rain corrosion damage constitutive relationship can be illustrated by Equation (17).
(16)$$E_{n}=E_{0}(1-D_{n})$$
(17)$$\sigma_{n}=E_{0}(1-D_{n})\varepsilon_{n}$$

(2) According to the variation law of the elastic modulus for brick masonry after acid rain corrosion cycles, the damage evolution equation with the number of acid rain corrosion cycles was established. Similar to the damage evolution of concrete, the following assumptions were made before giving the damage evolution equation: (1) The initial damage value of the masonry was considered to be zero before acid rain corrosion. (2) The masonry corrosion damage was only a function of the number of corrosion cycles, which ignores the influence of other factors. (3) As the number of corrosion cycles increased, the damage value grew gradually and the damage was positive.

According to the basic theory of macroscopic phenomenological damage mechanics, the masonry corrosion damage variable $D_{n}$ is defined as follows.
(18)$$D_{n}=\frac{E_{0}-E_{n}}{E_{0}}$$

(3) During the uniaxial compression of masonry, macroscopic compressive strain is generated under the action of external compressive stress. According to the equilibrium condition of the pressure direction of the macroscopic unit, the following relationship can be obtained.
(19)$$\sigma =E_{0}\varepsilon (1-D_{c})$$
where $D_{c}$ is a mesoscopic damage variable caused by external pressure.

The state after acid rain corrosion is regarded as the first damage state, and the total damage state caused by the axial pressure after acid rain corrosion cycle is regarded as the second damage state. Again, based on the principle of equivalent strain, the constitutive relationship of the axial compression under an acid rain corrosion cycle is deduced as follows.
(20)$$\sigma =E_{n}(1-D_{c})\varepsilon =E_{0}(1-D_{n})(1-D_{c})\varepsilon =E_{0}(1-D_{k})\varepsilon$$
where $1-D_{k}=(1-D_{c})(1-D_{n})=1-D_{c}-D_{n}+D_{c}D_{n}$, $D_{k}$ is the total damage of brick masonry under the action of acid rain corrosion and axial compression. The damage caused by an acid rain corrosion cycle as well as the damage caused by axial compression show clear nonlinear characteristics.

(4) A masonry structure is composed of mortar, brick, and the interface. Additionally, the material damage strength for brick masonry obeys the Weibull Distribution from a mesoscopic perspective, which has been proven in Reference [37]. Therefore, the compression damage variable *D* obeys the Weibull statistical distribution, and it can be described by Equation (21).
(21)$$D=1-\exp\left[-{\left(\varepsilon /\varepsilon_{0}\right)}^{m}\right]$$
where $\varepsilon_{0}$ and m represent the scale parameter and the shape parameter, respectively.

The mesoscopic statistical damage model is used to describe the damage constitutive of brick masonry as follows.
(22)$$\sigma =E_{0}\cdot \exp\left[-{\left(\varepsilon /\varepsilon_{0}\right)}^{m}\right]\cdot \varepsilon$$

Deriving Equation (22) yields the following equation.
(23)$$d\sigma /d\varepsilon =E\cdot \exp\left\{\left[1-m{\left(\varepsilon /\varepsilon_{0}\right)}^{m}\right]\cdot \left[-{\left(\varepsilon /\varepsilon_{0}\right)}^{m}\right]\right\}$$

From the basic characteristics of the uniaxial compressive stress-strain curve of brick masonry, the peak strain $\varepsilon_{m}$ corresponding to the peak stress $\sigma_{m}$ can be obtained by Equation (22).
(24)$$\sigma_{m}/E_{0}\varepsilon_{m}=\exp\left[-{\left(\varepsilon_{m}/\varepsilon_{0}\right)}^{m}\right]$$

Taking the natural logarithm twice for Equation (24), the following relationship can be obtained.
(25)$$\ln\left[\ln\left(E_{0}\varepsilon_{m}/\sigma_{m}\right)\right]=m\cdot \ln\left(\varepsilon_{m}/\varepsilon_{0}\right)$$

The slope of the peak point for the stress-strain curve is zero and the equation dσ/dε=0 has a unique non-zero solution, which indicates that there is only one peak on the curve and there is a maximum point. Thus, the following equation can be obtained by Equation (23).
(26)$$1/m={\left(\varepsilon_{m}/\varepsilon_{0}\right)}^{m}$$

Taking the natural logarithm of both sides, it can be determined that
(27)$$\ln\left(1/m\right)=m\ln\left(\varepsilon_{m}/\varepsilon_{0}\right)$$

For the simultaneous Equations (25) and (27), the expressions of the shape parameters are as follows.
(28)$$m=\ln^{-1}\left(E_{0}\varepsilon_{m}/\sigma_{m}\right)$$

The expression of the scale parameter can be obtained from Equation (26).
(29)$$\varepsilon_{0}=\varepsilon_{m}/{\left(m^{-1}\right)}^{m^{-1}}$$

Lastly, the masonry damage model can be obtained as follows.
(30)$$D=1-\exp\left[-\frac{1}{m}{\left(\frac{\varepsilon }{\varepsilon_{0}}\right)}^{m}\right]$$

(5) Based on Equations (18) and (30), the equation of the total damage evolution of brick masonry under uniaxial compression after acid rain corrosion is as follows.
(31)$$D_{k}=1-E_{n}/E_{0}\exp\left[-1/m{\left(\varepsilon /\varepsilon_{0}\right)}^{m}\right]$$

Therefore, the damage constitutive relationship considering an acid rain corrosion cycle can be expressed as follows.
(32)$$\sigma =E_{n}\cdot \exp\left[-1/m{\left(\varepsilon /\varepsilon_{m}\right)}^{m}\right]\cdot \varepsilon$$

(6) Figure 19 illustrates the comparison between the deduced damage constitutive model and the experimental data of brick masonry under different acid rain corrosion cycles. As Figure 19 shows, it was indicated that the uniaxial compression damage constitutive model of brick masonry could objectively reflect the variation of the uniaxial compression performance of brick masonry under different acid rain corrosion cycles.

## 6. Conclusions

This study presented a systematic experimental study that was undertaken to investigate the mechanical properties of masonry experiencing acid rain corrosion damage, which were required for the modeling and assessment of existing masonry buildings. The masonry units were conventional solid clay bricks, and three typical mortar prisms (cement mortar, cement-lime mortar, and cement-fly ash mortar) were considered. The major findings of this study were as follows.
(1)In the initial stage of acid rain corrosion, a sanding phenomenon appeared on the surface of the mortar prism, and there were white crystal spots. In the middle stage of acid rain corrosion, the surface color of the mortar prism changed, and the phenomenon of “skinning” occurred. At the end of the corrosion, the “skin” phenomenon on the surface of the mortar prism was aggravated and peeling occurred in some places.(2)The compressive strength of the mortar prism first increased and then decreased with the increase of the number of acid rain corrosion cycles. The compressive strengths of the cement-lime mortar, cement mortar, and cement-fly ash mortars increased by 10.42%, 9.78%, and 14.81% after 100 acid rain corrosion cycles, respectively. The compressive strengths of the cement-lime mortar prism, cement-fly ash mortar prism, cement mortar prism, and brick decreased by 24.79%, 15.30%, 19.01%, and 20.22% after 300 corrosion cycles, respectively. The cement-lime mortar prism had the worst corrosion resistance. The incorporation of fly ash into the cement mortar did not improve the acid rain corrosion resistance.(3)The acid rain environment had a great influence on the shear strength and compressive strength of the brick masonry. With the increasing number of acid rain corrosion cycles, the shear strength and the compressive strength of the brick masonry showed a trend of first increasing and then decreasing. After 300 acid rain corrosion cycles, the shear strength of the brick masonry specimens decreased by 13.1%, the compressive strength decreased by 29%, and the elastic modulus decreased by 42%. As the amount of corrosion increased, the peak stress decreased, whereas the peak strain increased and the slope of the stress-strain curve gradually declined.(4)Extrapolating the results to existing masonry buildings would require additional numerical and experimental investigations. The next phase of the research will focus on the mechanical properties (the compressive strength of mortar, the shear strength of brick masonry, the compressive strength of brick masonry, and the constitutive model of brick masonry) under different service ages and numerical modeling analysis of multi-age masonry structures in acidic atmospheric environment.

## Figures and Tables

**Figure 1 materials-12-02694-f001:**
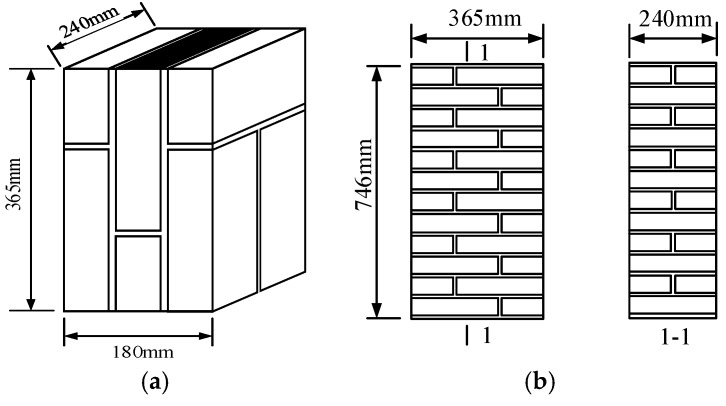
Layout of specimens used for (**a**) masonry direct shear and (**b**) masonry compression.

**Figure 2 materials-12-02694-f002:**
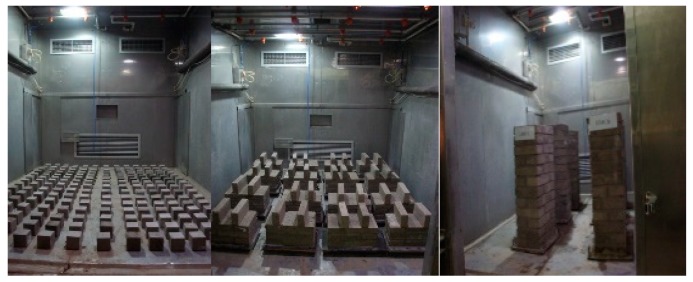
Specimens in laboratory exposure chambers.

**Figure 3 materials-12-02694-f003:**
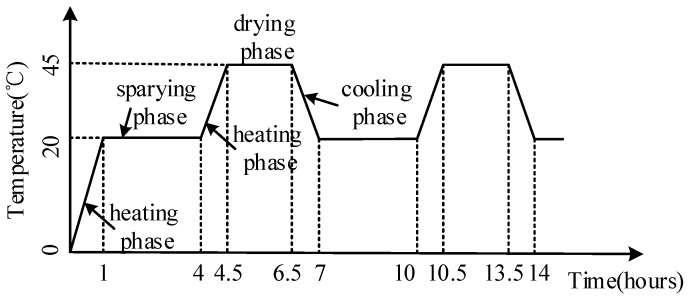
Corrosion cycle system schematic diagram.

**Figure 4 materials-12-02694-f004:**
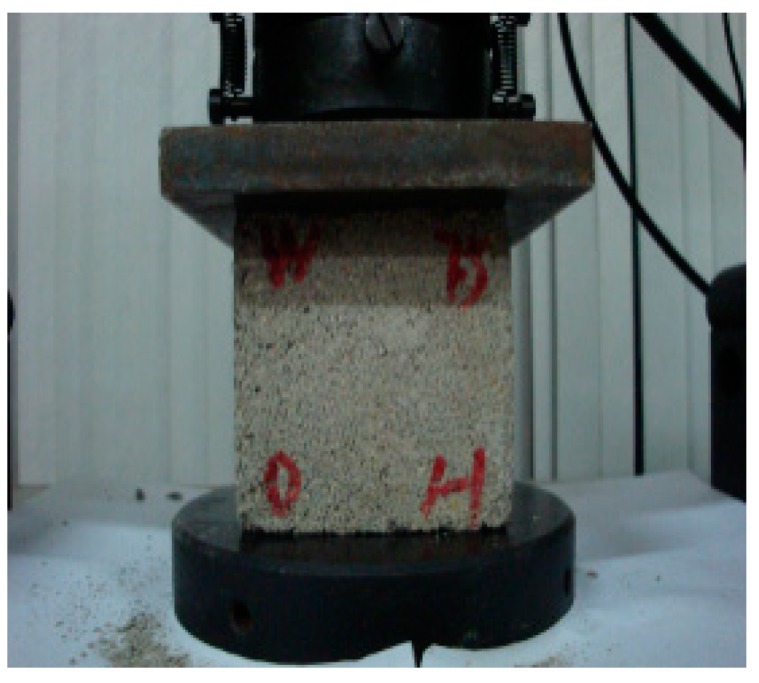
Compression test of mortar prism.

**Figure 5 materials-12-02694-f005:**
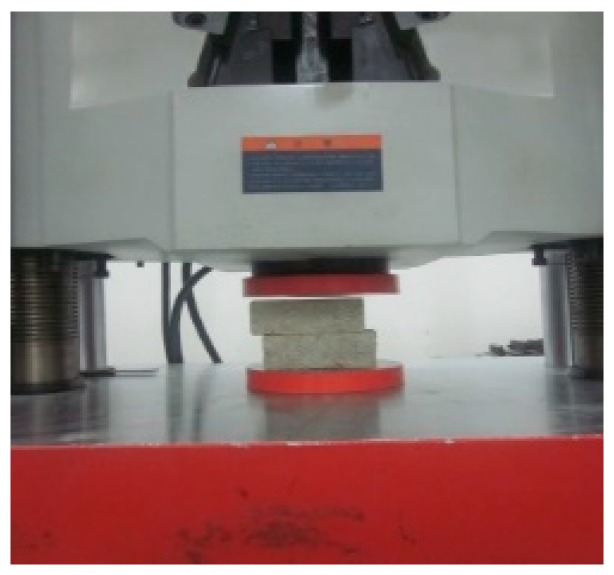
Compression test of brick.

**Figure 6 materials-12-02694-f006:**
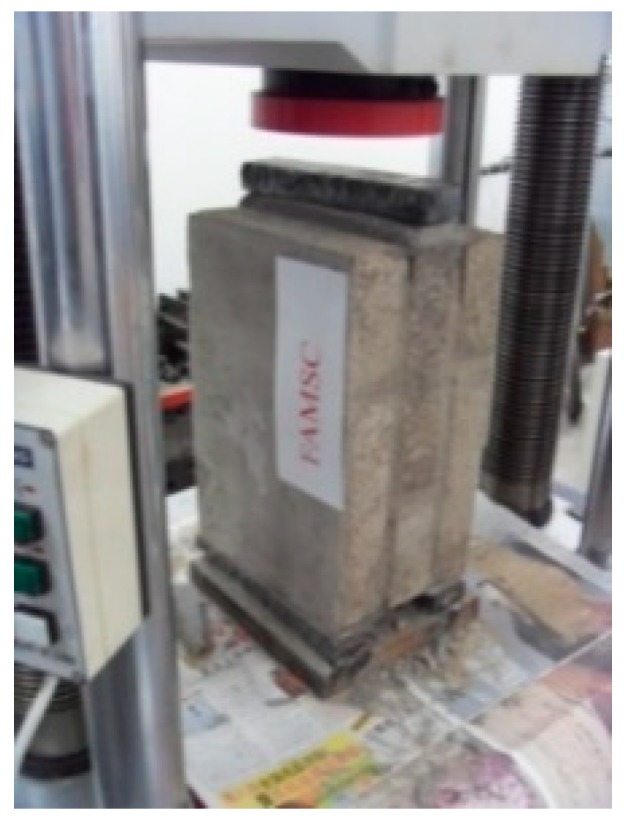
The test loading of the shear specimen.

**Figure 7 materials-12-02694-f007:**
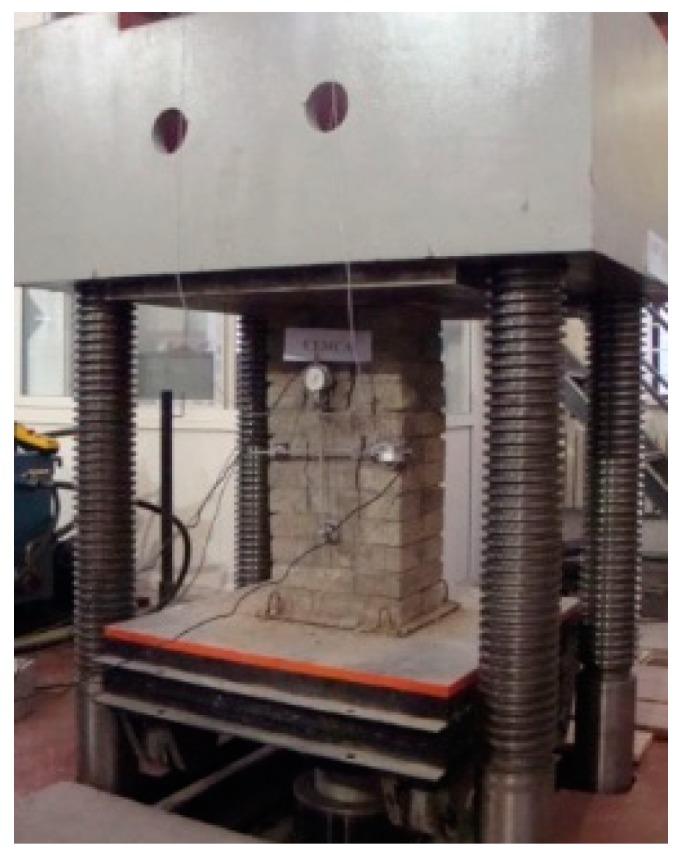
The test loading of the compressive specimen.

**Figure 8 materials-12-02694-f008:**
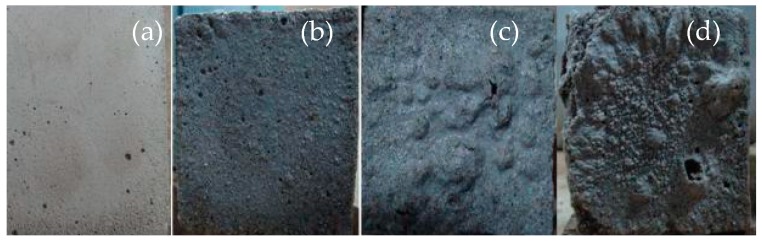
Apparent morphology of the mortar prism under different corrosion cycles. (**a**) 0, (**b**) 100 cycles, (**c**) 200 cycles, and (**d**) 300 cycles.

**Figure 9 materials-12-02694-f009:**
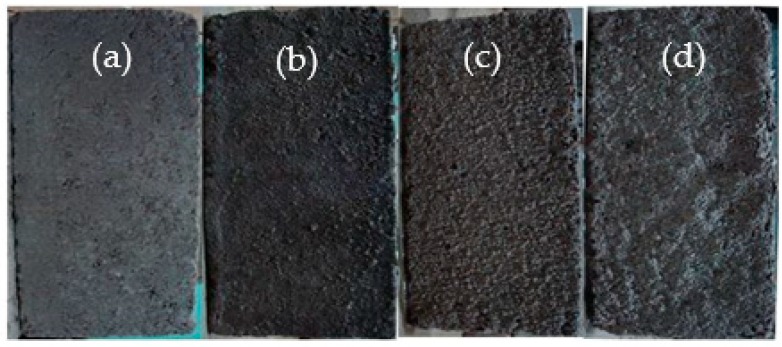
Apparent morphology of brick under different corrosion cycles. (**a**) 0, (**b**) 100 cycles, (**c**) 200 cycles, and (**d**) 300 cycles.

**Figure 10 materials-12-02694-f010:**
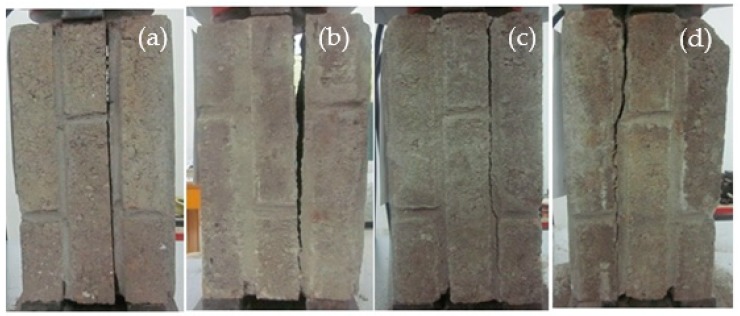
Failure pattern of shear specimen under different corrosion cycles. (**a**) KJMA. (**b**) KJMB. (**c**) KJMC. (**d**) KJMD.

**Figure 11 materials-12-02694-f011:**
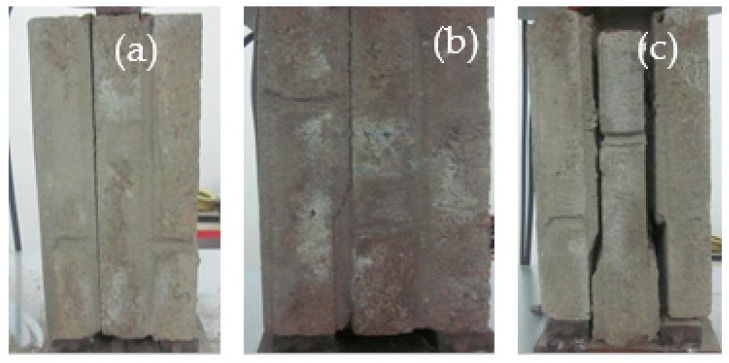
Failure pattern observed in shear test. (**a**) Single shear (0° angle). (**b**) Single shear (45° angle). (**c**) Double shear.

**Figure 12 materials-12-02694-f012:**
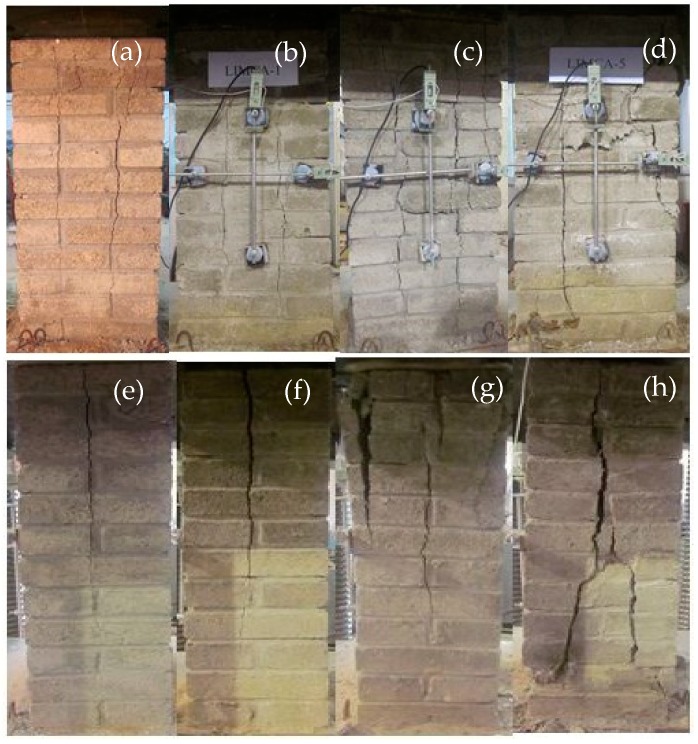
Failure pattern of compression specimen under a different corrosion cycle. (**a**) Wide side of KYMA. (**b**) Wide side of KYMB. (**c**) Wide side of KYMC. (**d**) Wide side of KYMD. (**e**) Narrow side of KYMA. (**f**) Narrow side of KYMB. (**g**) Narrow side of KYMC. (**h**) Narrow side of KYMD.

**Figure 13 materials-12-02694-f013:**
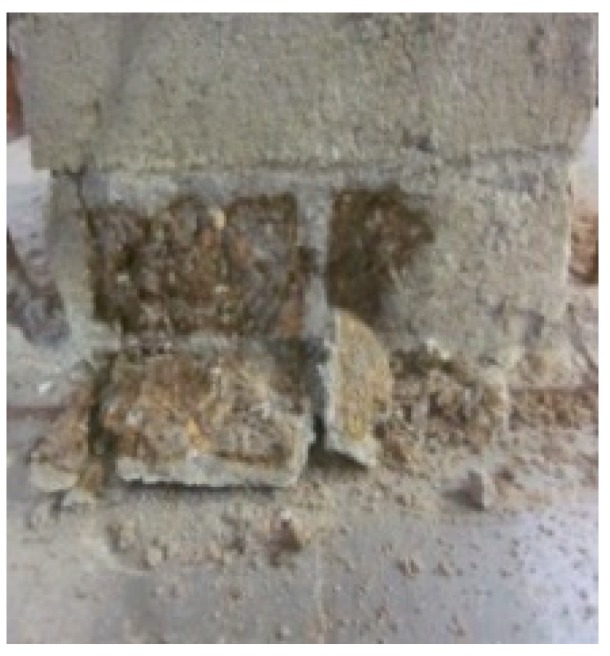
The feet of the corrosion specimen.

**Figure 14 materials-12-02694-f014:**
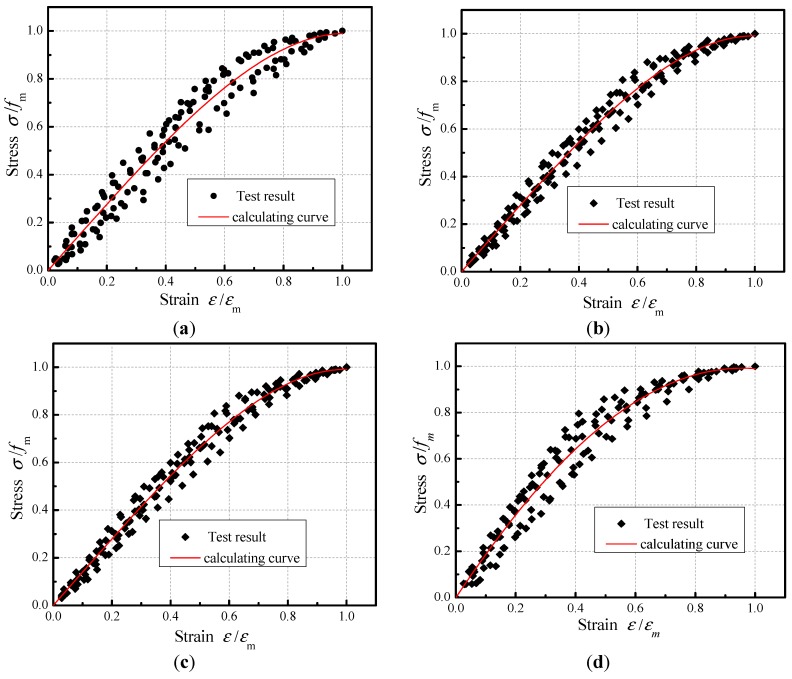
Fitting curves of stress-strain under different acid rain corrosion cycles. (**a**) 0 cycle, (**b**) 100 cycles, (**c**) 200 cycles, and (**d**) 300 cycles.

**Figure 15 materials-12-02694-f015:**
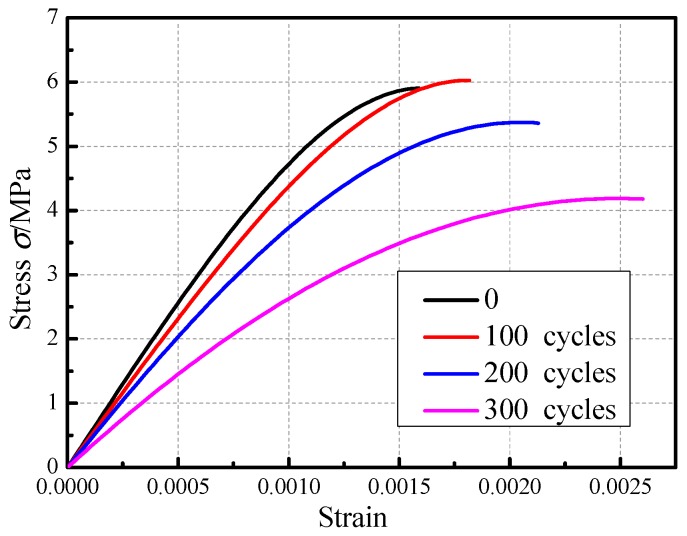
Stress–strain curve of the masonry prism subjected to different corrosion cycles.

**Figure 16 materials-12-02694-f016:**
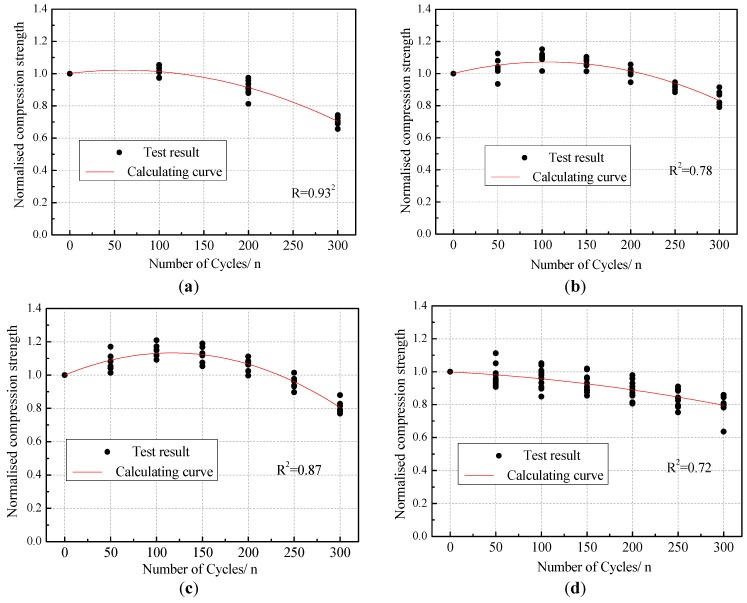
The compressive strength attenuation model of mortars and brick. (**a**) Cement-lime mortar prism. (**b**) Cement mortar prism. (**c**) Cement-fly ash mortar prism. (**d**) Brick unit.

**Figure 17 materials-12-02694-f017:**
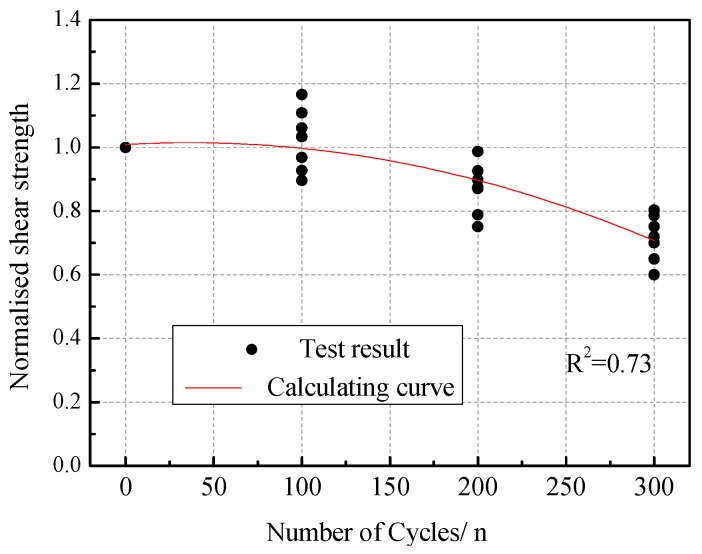
Shear strength attenuation model.

**Figure 18 materials-12-02694-f018:**
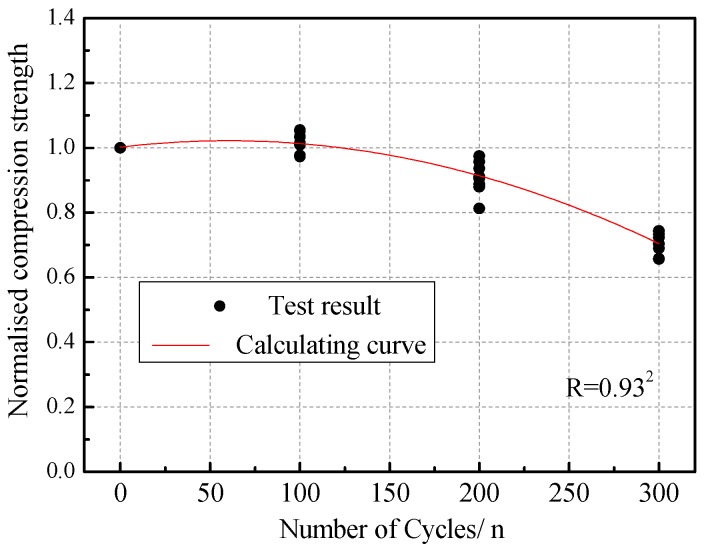
Compressive strength attenuation model.

**Figure 19 materials-12-02694-f019:**
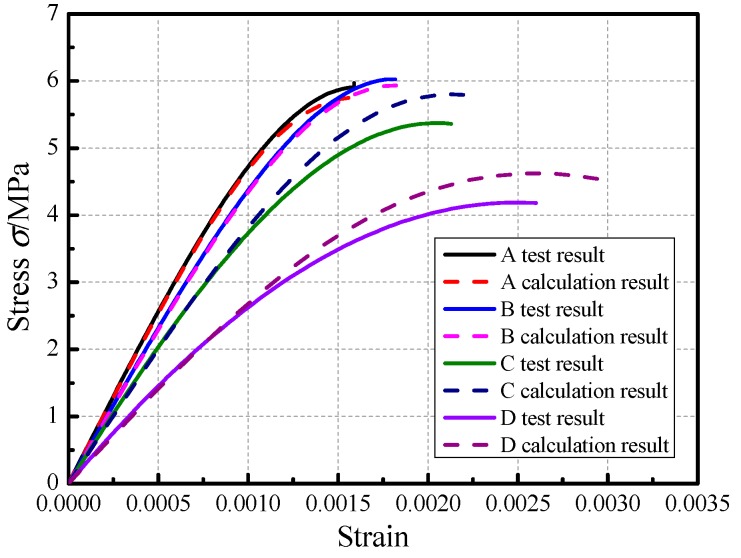
Comparison between proposed analytical models and experimental data for the specimens.

**Table 1 materials-12-02694-t001:** Main physical characteristics of cement.

Test Projects	Value
Initial setting time/min	80
Final setting time/min	300
Flexural strength/MPa	5.8
Compressive strength/MPa	34.6
Main components/%	SiO_2_ = 20.78, Fe_2_O_3_ = 4.44, Al_2_O_3_ = 6.18, CaO = 65.82, MgO = 1.92

**Table 2 materials-12-02694-t002:** Performance index of fly ash.

Test Projects	Value
Ignition loss/%	1.18–5.16
0.08 mm sieve residue/%	7.46
Main components/%	SiO_2_ = 52.2, Fe_2_O_3_ = 5.78, CaO = 7.32, Al_2_O_3_ = 22.6, MgO = 2.78, SO_3_ = 0.48

**Table 3 materials-12-02694-t003:** Performance index of brick.

Test Projects	Value
Compressive strength/MPa	15.2
Flexural strength/MPa	7.2
Density/kg/m^3^	2630
Drying shrinkage/%	0.08
Water absorption/%	12.3

**Table 4 materials-12-02694-t004:** Mix ratio of mortar.

Mortar Type	Cement/kg/m^3^	River Sand/kg/m^3^	Lime Past/kg/m^3^	Water/kg/m^3^	Fly Ash/kg/m^3^
CEM	275	1450	-	120	-
CEM-LIM	290	1450	90 (30%)	130	-
CEM-FLY	240	1450	-	105	102 (30%)

**Table 5 materials-12-02694-t005:** Number of samples and testing conditions for the samples.

Specimen	Test Type	Mortar Type	Group	Each Group
Brick	Compression	-	7	10
Mortar prism	Compression	CEM,CEM-LIM,CEM-LIM	7	6
Masonry	Direct shear	CEM-LIM	KJMA~KJMD	7
	Compression	CEM-LIM	KYMA~KYMD	8

**Table 6 materials-12-02694-t006:** Index properties for mortar prisms and brick.

Cycle Index/*n*	CEM-LIM			CEM			CEM-FLY			Brick		
	Mean/MPa	S d	LR/%	Mean/MPa	S d	LR/%	Mean/MPa	S d	LR/%	Mean/MPa	S d	LR/%
0	10.65	0.61	0.00	12.68	0.80	0.00	10.26	0.50	0.00	16.37	0.80	0.00
50	11.18	1.07	−4.98	13.15	0.74	−3.71	11.06	0.52	−7.80	15.96	0.98	2.50
100	11.76	0.82	−10.42	13.92	0.53	−9.78	11.78	0.38	−14.81	15.74	1.01	3.85
150	11.65	0.67	−9.39	13.51	0.37	−6.55	11.51	0.49	−12.18	15.22	0.89	7.03
200	10.9	0.67	−2.35	12.78	0.43	−0.79	10.86	0.39	−5.85	14.52	0.90	11.30
250	9.58	0.58	10.05	11.63	0.27	8.28	9.78	0.38	4.68	13.78	0.82	15.82
300	8.01	0.46	24.79	10.74	0.57	15.30	8.31	0.37	19.01	13.06	0.99	20.22

Note: Mean is compressive strength, S d is Standard deviation, and LR is the strength loss rate.

**Table 7 materials-12-02694-t007:** Shear strength of masonry under different acid rain corrosion cycles.

Group	Cycle Index/*n*	Specimen Code	$N_{\mathrm{vu}}{}_{,i}^{}/\mathrm{kN}$	$f_{\mathrm{vm},i}/\mathrm{MPa}$	$f_{\mathrm{vm}}/\mathrm{MPa}$	Standard Deviation	Failure Mode
KJMA	0	KJMA1	116	0.669	0.668	0.04	single shear
		KJMA2	125	0.722			double shear
		KJMA3	106	0.622			single shear
		KJMA4	107	0.599			single shear
		KJMA5	119	0.679			single shear
		KJMA6	120	0.700			double shear
		KJMA7	120	0.688			single shear
KJMB	100	KJMB1	121	0.691	0.684	0.06	single shear
		KJMB2	127	0.741			double shear
		KJMB3	136	0.779			double shear
		KJMB4	109	0.620			single shear
		KJMB5	105	0.599			single shear
		KJMB6	113	0.647			single shear
		KJMB7	124	0.709			double shear
KJMC	200	KJMC1	100	0.584	0.582	0.05	single shear
		KJMC2	108	0.619			double shear
		KJMC3	116	0.660			double shear
		KJMC4	88	0.502			single shear
		KJMC5	92	0.527			single shear
		KJMC6	105	0.600			single shear
		KJMC7	102	0.582			single shear
KJMD	300	KJMD1	84	0.481	0.478	0.05	single shear
		KJMD2	94	0.537			double shear
		KJMD3	91	0.525			single shear
		KJMD4	81	0.468			single shear
		KJMD5	74	0.434			single shear
		KJMD6	70	0.401			single shear
		KJMD7	86	0.502			single shear

**Table 8 materials-12-02694-t008:** Comparison of average compressive strength between calculated values and measured values.

Group	Cycle Index/*n*	Specimen Code	$N_{\mathrm{cr},i}/\mathrm{kN}$	$N_{\mathrm{cu},i}/\mathrm{kN}$	$f_{\mathrm{cm},i}/\mathrm{MPa}$	$f_{\mathrm{cm}}/\mathrm{MPa}$	Standard Deviation	$f_{m}^{c}/\mathrm{MPa}$	RCS1	RCS2
KYMA	0	KYMA1	488	543	6.194	5.971	0.40	5.497	1.000	1.000
		KYMA2	443	515	6.005					
		KYMA3	467	556	6.506					
		KYMA4	417	521	5.850					
		KYMA5	401	572	6.571					
		KYMA6	427	486	5.559					
		KYMA7	384	492	5.627					
		KYMA8	381	465	5.459					
KYMB	100	KYMB1	358	498	5.816	6.082	0.17	5.642	1.019	1.026
		KYMB2	462	538	6.297					
		KYMB3	388	498	5.833					
		KYMB4	484	550	6.176					
		KYMB5	446	525	6.022					
		KYMB6	439	549	6.279					
		KYMB7	415	526	6.176					
		KYMB8	433	528	6.056					
KYMC	200	KYMC1	366	489	5.589	5.424	0.28	5.240	0.908	0.953
		KYMC2	330	459	5.252					
		KYMC3	334	464	5.443					
		KYMC4	343	490	5.721					
		KYMC5	298	432	4.856					
		KYMC6	385	507	5.824					
		KYMC7	354	454	5.296					
		KYMC8	342	462	5.413					
KYMD	300	KYMD1	269	374	4.378	4.218	0.20	4.400	0.706	0.800
		KYMD2	300	395	4.439					
		KYMD3	251	342	3.927					
		KYMD4	235	335	3.927					
		KYMD5	254	374	4.195					
		KYMD6	256	376	4.317					
		KYMD7	234	360	4.122					
		KYMD8	249	378	4.439					

Note: $N_{\mathrm{cr},i}$ is the crack load of compression specimen, $N_{\mathrm{cu},i}$ is the ultimate load of compression specimen, $f_{\mathrm{cm},i}$ is the measured compressive strength, $f_{\mathrm{cm}}$ is the average measured compressive strength, $f_{m}^{c}$ is the calculated compressive strength.

**Table 9 materials-12-02694-t009:** Fitting parameters of stress-strain curves.

Cycle Index/*n*	0	100	200	300
A	1.39	1.4	1.66	1.93
B	0.4	0.41	0.67	0.94
C	3.37	3.43	2.63	2.16
*R* ^2^	0.97	0.98	0.98	0.97
*f_m_*/MPa	5.97	6.08	5.42	4.22
$\varepsilon_{m}$/10^−3^	1.589	1.817	2.129	2.602

Note: *R*^2^ is the correlation coefficient.

**Table 10 materials-12-02694-t010:** Elastic modulus of the specimen.

Cycle Index/*n*	0	100	200	300
**Elastic Modulus/MPa**	5012	4736	4038	2918

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
