# Peer review of "Mechanical Behavior of Brick Masonry in an Acidic Atmospheric Environment"

_materials, 2019, doi:10.3390/ma12172694_

Round 1

Reviewer 1 Report

The authors present a research work related to “Mechanical behavior of brick masonry in an acidic atmospheric environment” where a systematic experimental study was undertaken to investigate the mechanical properties of masonry experiencing acid rain corrosion damage.

Remarks to the authors:

1. Under Table 2 “The size of the brick was 240X115X53 mm3

Dimension (L*W*H): 240x115x53 mm

2. Figures and tables should be placed in the main text near to the first time they are cited.

3. What would the authors suggest based on the results?

4. Future research directions should be mentioned.

Author Response

1. Thanks for the referee’s kind suggestion. According to his/her advices, the expression of brick dimension were revised.

2. According to his/her advices, figures and tables was placed in the main text near to the first time they are cited.

3. (1) The quantitative relationship between the durability damage degree of multi-age masonry structure and the service age in the general (acidic) atmosphere environment was established by our research team. Combined with the research results of this paper, the mechanical properties (the compressive strength of mortar, the shear strength of brick masonry, the compressive strength of brick masonry and the stress–strain curve of brick masonry) under different service ages can be obtained.

Considering the length of this paper, the quantitative relationship between the durability damage degree of multi-age masonry structure and the service age in the general (acidic) atmosphere environment will be published in another article.

   (2) In this paper, the mechanical properties of brick masonry experiencing acid rain corrosion damage were studied, and the deterioration law of the mechanical properties (the compressive strength of mortar, the shear strength of brick masonry, the compressive strength of brick masonry and the constitutive structure of brick masonry) of brick masonry after acid rain corrosion were established. The research results can provide theoretical basis for numerical modeling analysis of multi-age masonry structures in general (acidic) atmospheric environment.

4. (1) Future research directions were brief described as follows:

The paper obtained the degradation law of mechanical properties of brick masonry under different acid rain corrosion cycles. At the same time, the quantitative relationship between the durability damage degree of multi-age masonry structure and the service age in the general (acidic) atmosphere environment was established. Then, the mechanical properties of brick masonry of different service ages under acidic atmospheric environment can be obtained. Furthermore, the research results also can provide theoretical basis for numerical modeling analysis of multi-age masonry structures in general (acidic) atmospheric environment.

(2)Future research directions were add in Section 6.

Extrapolating the results to existing masonry buildings would require additional numerical and experimental investigations. The next phase of the research will focus on the mechanical properties (the compressive strength of mortar, the shear strength of brick masonry, the compressive strength of brick masonry and the constitutive model of brick masonry) under different service ages and numerical modeling analysis of multi-age masonry structures in acidic atmospheric environment.

Reviewer 2 Report

materials-560411

Title: Mechanical behavior of brick masonry in an acidic atmospheric environment.

The paper deals with acid corrosion of brick masonry. Simulated acid rain corrosion tests and mechanical tests on mortars, bricks and shear prisms were conducted. Furthermore, degradation models and damage constitutive relation model for masonry were established.

After a thorough read, I can say that: a) Abstract is not concise and informative; b) Introduction, methods, collected data and conclusions are of good level; c) Measurement units are not always correctly written. Therefore, I recommend the paper publication after revision of Abstract sentences and of typing errors.

Author Response

a) Abstract is not concise and informative

Abstract were revised.

In order to evaluate the deterioration regularity for the mechanical properties of brick masonry due to acid rain corrosion, a series of mechanical property tests for mortars, bricks, shear prisms and compressive prisms after acid rain corrosion were conducted. The apparent morphology and the compressive strength of the masonry materials (cement mortar, cement-lime mortar, cement-fly ash mortar and brick), the shear behavior of the masonry, and the compression behavior of the masonry were analyzed. The resistance of acid rain corrosion for the cement-lime mortar prisms was the worst, and the incorporation of fly ash into the cement mortar did not improve the acid rain corrosion resistance. The effect of the acid rain corrosion damage on the mechanical properties for the brick was significant. With an increasing number of acid rain corrosion cycles, the compressive strength of the mortar prisms, and the shear and compressive strengths of the brick masonry first increased and then decreased; the peak stress first increased and then decreased whereas the peak strain gradually increased; the slope of the stress-strain curve for the compression prisms gradually decreased. Furthermore, a mathematical degradation model for the compressive strength of the masonry material (cement mortar, cement-lime mortar, cement-fly ash mortar and brick), as well as the shear strength attenuation model and the compressive strength attenuation model of brick masonry after acid rain corrosion were proposed. Finally, a damage constitutive relation model of masonry subjected to acid rain corrosion was deduced from the meso-damage theory.

c) Measurement units are not always correctly written.

The measurement units were revised.